# Identification of Survival-Specific Genes in Clear Cell Renal Cell Carcinoma Using a Customized Next-Generation Sequencing Gene Panel

**DOI:** 10.3390/jpm12010113

**Published:** 2022-01-14

**Authors:** Jia Hwang, Heeeun Kim, Jinseon Han, Jieun Lee, Sunghoo Hong, Saewoong Kim, Sungjoo Kim Yoon, Keonwoo Choi, Jihoon Yang, Unsang Park, Kwangjoong Kim, Kwangil Yim, Yuil Kim, Yeongjin Choi

**Affiliations:** 1Department of Hospital Pathology, Seoul St. Mary’s Hospital, College of Medicine, The Catholic University of Korea, Seoul 06591, Korea; hja0329@hotmail.com (J.H.); heya0808@gmail.com (H.K.); jinseon-han@hotmail.com (J.H.); jning@naver.com (J.L.); kangse_manse@catholic.ac.kr (K.Y.); yuil.gim@gmail.com (Y.K.); 2Department of Urology, Seoul St. Mary’s Hospital, College of Medicine, The Catholic University of Korea, Seoul 06591, Korea; toomey@catholic.ac.kr (S.H.); ksw1227@catholic.ac.kr (S.K.); 3Department of Biomedicine and Health Sciences, College of Medicine, The Catholic University of Korea, Seoul 06591, Korea; sjkyoon@catholic.ac.kr (S.K.Y.); icsoo123@naver.com (K.C.); 4Department of Computer Science and Engineering, Sogang University, Seoul 04107, Korea; yangjh@sogang.ac.kr (J.Y.); unsangpark@sogang.ac.kr (U.P.); 5NGeneBio Co., Ltd., Seoul 08390, Korea; kjoong.kim@ngenebio.com

**Keywords:** clear cell renal cell carcinoma, survival, genes, mutations, *CARD6*, NGS, Korea

## Abstract

Purpose: Although mutations are associated with carcinogenesis, little is known about survival-specific genes in clear cell renal cell carcinoma (ccRCC). We developed a customized next-generation sequencing (NGS) gene panel with 156 genes. The purpose of this study was to investigate whether the survival-specific genes we found were present in Korean ccRCC patients, and their association with clinicopathological findings. Materials and Methods: DNA was extracted from the formalin-fixed, paraffin-embedded tissue of 22 ccRCC patients. NGS was performed using our survival-specific gene panel with an Illumina MiSeq. We analyzed NGS data and the correlations between mutations and clinicopathological findings and also compared them with data from the Cancer Genome Atlas-Kidney Renal Clear Cell Carcinoma (TCGA-KIRC) and Renal Cell Cancer-European Union (RECA-EU). Results: We found a total of 100 mutations in 37 of the 156 genes (23.7%) in 22 ccRCC patients. Of the 37 mutated genes, 11 were identified as clinicopathologically significant. Six were novel survival-specific genes (*ADAMTS10*, *CARD6*, *NLRP2*, *OBSCN*, *SECISBP2L*, and *USP40*), and five were top-ranked mutated genes (*AKAP9*, *ARID1A*, *BAP1*, *KDM5C*, and *SETD2*). Only *CARD6* was validated as an overall survival-specific gene in this Korean study (*p* = 0.04, *r* = −0.441), TCGA-KIRC cohort (*p* = 0.0003), RECA-EU (*p* = 0.0005). The 10 remaining gene mutations were associated with clinicopathological findings; disease-free survival, mortality, nuclear grade, sarcomatoid component, N-stage, sex, and tumor size. Conclusions: We discovered 11 survival-specific genes in ccRCC using data from TCGA-KIRC, RECA-EU, and Korean patients. We are the first to find a correlation between CARD6 and overall survival in ccRCC. The 11 genes, including *CARD6*, *NLRP2*, *OBSCN*, and *USP40*, could be useful diagnostic, prognostic, and therapeutic markers in ccRCC.

## 1. Introduction

Clear cell renal cell carcinoma (ccRCC) is one of the most lethal cancer types globally, accounting for 70–75% of all renal cancers [1]. In South Korea, ccRCC accounted for approximately 85% of all renal cell carcinoma (RCC) cases in 2015 [2]. Several studies on ccRCC-related mutant genes have been reported based on next-generation sequencing (NGS) techniques, but most of them were performed on a small scale [3].

The clinical, morphological, and genetic characteristics of RCC have been found to vary significantly from patient to patient due to high tumor heterogeneity [4]. It has been widely known that cancer is caused by the accumulation of cancer-causing gene mutations, and in fact, at least one protein-altering mutation is found in 85% of cancer tissues [5]. Therefore, a diverse clinical picture arises due to the diversity of genetic mutations.

Recently, various genetic mutations causing RCC have been identified by high-throughput sequencing methods [6,7,8,9,10], such as those used in The Cancer Genome Atlas Kidney Renal Clear Cell Carcinoma (TCGA-KIRC) and Renal Cell Cancer-EU/FR (RECA-EU) databases, and many predisposing genes involved in the development of RCC have been reported through hereditary renal cancer studies [6,8,9,10,11]. However, studies on the most important gene mutations related to the survival of RCC patients are lacking, and most genetic studies are based on databases from Western populations. Few mutation data are based on the East Asian population.

Currently, NGS, which analyzes multiple genes in parallel at the same time, is the most useful method for diagnosing gene mutations [12,13]. To date, little is known about genetic markers that predict survival in patients with ccRCC. However, the application of personalized therapy requires the development of dedicated NGS panels for key target genes associated with important clinicopathological conditions, including survival.

In this study, we developed a customized NGS gene panel for ccRCC using 156 genes. This panel included 123 survival-specific genes newly discovered by machine learning, and 21 of the top-ranked mutated genes above 5% in TCGA-KIRC, as well as genes from other databases. We used this panel to determine whether survival-specific gene mutations are present in Korean ccRCC patients and to analyze their association with clinicopathological findings. We discovered and validated 11 survival-specific genes in ccRCC patients based on TCGA-KIRC and RECA-EU data as well as data from Korean patients. To our knowledge, we are the first group to report a correlation between *CARD6* mutation and overall survival (OS) in patients with ccRCC. We believe that the 11 genes, especially *CARD6*, *NLRP2*, *OBSCN*, and *USP40*, could be used as useful diagnostic, prognostic, and therapeutic markers in ccRCC.

## 2. Materials and Methods

### 2.1. Ethical Statement

All procedures performed in this study were in accordance with the 1964 Helsinki declaration and its later amendments or comparable ethical standards, and with the ethical standards approved by the Institutional Review Board of Catholic University of Korea, Seoul St. Mary’s Hospital (approval no. 2018-2550-0008, date of approval: 25 September 2020). The retrospective genetic study and the treatment plan were conducted according to clinical guidelines and standard of care. The present genetic study results did not affect the treatment plan of patients following surgery. Informed written consent was provided by all patients.

### 2.2. Machine Learning and Statistical Methods for the Discovery of 123 Survival-Specific Genes

We conducted machine learning (ML) using Rapidminer (7.3 version, Boston, MA, USA) to discover survival-specific mutations in ccRCC from the TCGA database. From TCGA-KIRC, 417 patients with both somatic nonsilent mutations and clinical-related data were selected. The clinical data were obtained from the TCGA portal, and the Mutation Annotation Formatted file was obtained from UCSC Xena (UCSC, https://xenabrowser.net/). UCSC Xena data provided clinical information on mutations in 39,532 genes of 417 patients. ML techniques of three feature selection methods (Information Gain, Chi-squared test, and Minimum Redundancy Maximum Relevance (MRMR)) and three classifiers (Naïve Bayes, K-Nearest Neighbor (K-NN), and Support Vector Machine (SVM)) were implemented. The performances of the used models, shown in Appendix A, were compared and analyzed using survival graphs of the top 100 mutant genes selected by each model (Appendix A). The best-performing ML models were repeatedly utilized, and a total of 195 mutant genes related to the six clinicopathologic factors (age, sex, stage, recurrence, metastasis, and survival) were selected. To increase the accuracy, only genes commonly found by the three methods were extracted. Of 195 genes, 123 statistically validated genes were selected by analysis of variance (ANOVA) and a Fisher’s exact test after analyzing their mutation frequencies and overall survival (OS) and disease-free survival (DFS) (Appendix A).

### 2.3. Patients

Patients were enrolled from Catholic University College of Medicine, Seoul, Republic of Korea. The inclusion criteria of the study were patients who received radical/partial nephrectomy and pathology diagnosis of ccRCC and were willing to participate and provide signed informed consent. All patients with a renal tumor diagnosis underwent radical nephrectomy according to clinical indications. The pathology of each tumor was reviewed by pathologists specializing in kidney cancer identification, and only those diagnosed as ccRCC were included in the study. We selected 22 patients with ccRCC for this study. Clinicopathological factors of sex, age, tumor size, nuclear grade, sarcomatoid component, TNM stage (T = extent of the primary tumor, N = regional lymph nodes, M = distant metastases), and response to laparoscopic radical nephrectomy (LRN), and their results, are summarized in Table 1. Clinical demographic parameters; cancer stage using the American Joint Committee on Cancer; and pathological data of tumor stage, lymph node status, and Fuhrman nuclear grade were collected.

### 2.4. Samples

The biospecimens for this study were provided by the Biobank of Seoul St. Mary’s Hospital, the Catholic University of Korea. The formalin-fixed, paraffin-embedded (FFPE) samples of 22 paired normal and tumor tissues were obtained from ccRCC patients.

### 2.5. Customized NGS Gene Panel

We developed a customized NGS gene panel for ccRCC with 156 genes. This panel included 123 survival-specific genes newly discovered by machine learning, 21 of the mutated genes ranked above 5% in TCGA-KIRC (Appendix A), and 14 solid tumor-associated genes (Appendix A).

### 2.6. Library Preparation and Sequencing

Genomic DNA was extracted from FFPE tissues using a GeneRead DNA FFPE Kit (Cat. No. 180134, Qiagen, Hilden, Germany) according to the manufacturer’s instructions. Targeted DNA sequencing libraries were constructed using a SOLIDaccutest–renal cancer panel (Version 1.1, NGeneBio, Seoul, Korea) with 400 ng of genomic DNA. Briefly, genomic DNA was fragmented and tagged to generate adapter-tagged libraries. Each library was indexed by sample and pooled for target enrichment. Target enrichment was performed by hybridization capture using a custom-designed gene panel that targeted the entire coding sequence (CDS, 804 Kb) of 156 genes specific to ccRCC. The quality of the libraries was measured with TapeStation 4200 (Version A.02.02., Agilent Technologies, Santa Clara, CA, USA) and quantified using Qubit 4 (Thermo Fisher Scientific, Waltham, MA, USA). Sequencing was performed on an Illumina MiSeq and Nextseq platform (Version 3.1 and Version 4.0, Illumina, San Diego, CA, USA).

### 2.7. NGS Data Analysis

DNA sequencing data were analyzed using a SOLIDaccutest–renal cancer panel customized NGS analysis pipeline (NGeneBio, Seoul, Korea). Briefly, FASTQ files that store the sequence fragments of each sample were aligned to the hg19 reference genome using an alignment algorithm for aligning sequence reads or assembly contigs against a large reference genome such as the human genome. After read mapping, various post-processes such as duplicate removal, base quality recalibration, and read sorting were performed following the best practices recommended by Genome Analysis Tool Kit (GATK4 (Version 4.0.12.0), Broad Institute, Cambridge, MA, USA) [14]. Single-nucleotide polymorphisms (SNPs) and indel variants were detected using GATK4 Mutect2 (Version 4.0.12.0, Broad Institute, Cambridge, MA, USA) [15], and the variants at Variant allele frequency >2%, alternate allele count >5, and depth >100 were included in this study. Variant annotation was performed by a combination of SnpEff and snpSift (Version 4.0, Pablo Cingolani, http://pcingola.github.io/SnpEff/, accessed on 15 November 2021) [16]. Nonpathogenic (population frequency >1%) polymorphisms were filtered out using large-scale genome databases such as gnomAD [17] and 1000 Genome [18]. Pathogenicity was determined by the in-house database constructed using databases provide genomic interpretations such as COSMIC [19], ClinVar [20], and various cancer-related literature. Genes that were interpreted as false positives due to poor mapping quality were excluded from the analysis.

### 2.8. Dataset

We gathered data from the Cancer Genome Atlas (TCGA) and Renal Cell Cancer-EU/FR (RECA-EU) projects for closer scrutiny. They encompass clinical features from each patient including demographics, tumor stage, vital status since the first surgical procedure, and corresponding sequencing reads from their cancer genome, regardless of the class of variant. The TCGA-KIRC dataset can be found at https://www.cbioportal.org/study/summary?id=kirc_tcga (accessed on 7 April 2021), and the renal cell cancer (EU/FR) data from the RECA-EU are available for download at https://dcc.icgc.org/projects/RECA-EU (accessed on 27 May 2021). Both samples originated from renal cell carcinoma, and 451 of 538 donors from TCGA-KIRC and 422 of 475 donors from RECA-EU were selected from donors with simple somatic mutation to optimize the quality of data analysis. The TCGA-KIRC dataset covers clear cell carcinoma only, whereas the RECA-EU data focus on but are not limited to the clear cell subtype. They both use hg19 (GrCh37) annotation. To know the mutation frequency in the normal control population in South Korea, we also had access to data from the Korea Biobank Array Project (referred to as KoreanChip), approved by the National Biobank of Korea, the Centers for Disease Control and Prevention, Republic of Korea (KBN-2019-019, approval date: 21 March 2019). The Korea Biobank Array Project was initiated in 2014 by the Korea National Institute of Health and included 210,000 participants aged 40–69 years via the Korean Genome and Epidemiology Study [21] to implement a customized Korean genome structure-based array with high genomic coverage and abundant functional variants of low to rare frequency [22]. The KoreanChip comprised >833,000 markers including >247,000 rare-frequency or functional variants estimated from >2500 sequencing data in Koreans. Of the 833,000 markers, 208,000 functional markers were genotyped. Particularly, >89,000 markers were present in East Asians.

### 2.9. Data Preparation

For further clarification of our findings, we divided the donors obtained from TCGA-KIRC and RECA-EU datasets into two independent groups to exam the propensity for OS and DFS: the group with mutations in the selected genes most likely to contribute to renal cancer and the control group. To investigate the relationships between clinical characteristics and the mutated genes, we extracted data on six distinctive clinical variables: age, sex, tumor stage, metastasis, survival time, and recurrence. The categorical variables from clinical data were one-hot encoded; indicated by binary values, the rest were used as ordinal, and the unit of survival time was months. The raw data from the RECA-EU containing Ensembl IDs were converted to hg19 RefSeq annotation before mapping them into the gene list from the TCGA-KIRC. The raw format of KoreanChip data was processed using PLINK (Version 1.9, Shaun Purcell and Christopher Chang, www.cog-genomics.org/plink/1.9/, accessed on 15 November 2021). Genomic variants were genotyped from SNPs using Ensembl Variant Effect Predictor (VEP) (Version 96, Sarah E Hun (2019), http://apr2019.archive.ensembl.org/index.html, accessed on 15 November 2021), and additional identifiers were retrieved using the biomaRt package from R (Version 3.5, Durinck S, https://bioconductor.org/packages/release/bioc/html/biomaRt.html, accessed on 15 November 2021).

### 2.10. Statistical Analysis

To find the genes most strongly associated with renal cancer, a Pearson correlation analysis was performed to validate the associations between the clinical variables and the genes with clinicopathological importance. This method was also applied to evaluate the strength of association with the survival-related factors and to plot the correlation matrix table. The correlation coefficients, denoted by Pearson’s R, were measured by the function cor in the Stats R-Package (RStudio Team (2016), Boston, MA, USA, http://www.rstudio.com, accessed on 15 November 2021), and any coefficient greater than 0.3 was re-evaluated using Spearman’s test. When performing survival analyses using datasets from TCGA-KIRC and RECA-EU, the Kaplan–Meier method and the multivariate Cox proportional hazard model were used. The survival curves and hazard ratios over the given survival periods were estimated by utilizing a Python library called lifelines (0.26.4 version, Cameron Davidson-Pilon, https://lifelines.readthedocs.io/en/latest/). All pairs of survival curves were compared using the log-rank test module from lifeline statistics to confirm the discrepancy between groups. During the analysis, statistical significance was determined with *p* < 0.05 as a cutoff value and with a 95% confidence level.

## 3. Results

We developed a customized NGS gene panel of 156 genes related to ccRCC, including 123 survival-specific genes, and performed NGS analysis on samples from 22 patients. Of the 156 genes, 37 were mutated with a total of 100 mutations (data not shown). Of the 37 genes, 16 were in the top 5% or more, and 21 survival-specific genes were identified. The steps we performed, from gene discovery to NGS data analysis, are illustrated in Figure 1.

The clinicopathological findings of 22 ccRCC patients are summarized in Table 1. The patients were 15 men and seven women with a mean age of 59.5 years (33–82 years). Their mean tumor size was 6.6 cm (2.5–14.5 cm). The prevalence of nuclear grades I, II, III, and IV were 0 (0%), 6 (27.3%), 11 (50%), and 5 (22.7%), respectively. The prevalence of TNM stages was as follows. Tumor stage: seven patients (31.8%) with T1, five (22.7%) with T2, and 10 (45.5%) with T3. Lymph node staging showed 21 patients (95.5%) with N0 and one (4.5%) with N1. Distant metastasis staging indicated 12 patients (54.5%) with M0 and 10 (45.5%) with M1. Metastases were found in 10 of 22 (45.5%) ccRCC patients, all of which occurred in the lung and also in other sites such as the bones (3/22, 13.6%), brain (2/22, 9.1%), abdomen (1/22, 4.5%), liver (2/22, 9.1%), retroperitoneum (1/22, 4.5%), jejunum (1/22, 4.5%), and peritoneum (1/22, 4.5%). The mean OS of the patients was 34.6 months (12–50 months), and the mean DFS was 22.9 months (0–48 months). Of the 22 patients who underwent LRN, 12 (54.5%) showed no evidence of disease, and the remaining 10 (45.5%) had recurrence or metastasis due to treatment failure. Four of 22 patients (18.2%) died of ccRCC, and recurrence and metastasis were observed in all of them. The frequencies of the top 16 mutant genes found in this study are shown in Table 2. The mutation frequency of each gene in our study was compared with data from other studies, and the *VHL* gene was found to have the highest frequency. The frequencies of four genes (*VHL*, *SETD2*, *TSHZ3*, and *SPEN*) were particularly high (91%, 50%, 14%, and 18%, respectively) compared to other studies such as TCGA (54%, 21%, 6%, and 5%), RECA-EU (61%, 22%, 8%, and 5%), Tokyo (41%, 11%, 2.8%, and 0), and Taiwan (50%, 22%, unknown, and unknown) [10]. Our results also differed from other studies in terms of mutation frequency. In particular, mutations in *VHL* were found in 20 (91%) patients, which was noticeably high when compared to other results. All *VHL* gene mutations in 20 ccRCC patients were different, indicating that each tumor was highly heterogeneous genetically. Of the *VHL* mutations identified, 16 variants were reported to COSMIC, and the remaining four variants were not reported. In ClinVar, 10 variants were reported as pathogenic, two as likely pathogenic, one as of uncertain significance, and seven as unknown (Table 3).

The associations between the mutated genes in 22 Korean patients with ccRCC and their corresponding clinicopathological factors are shown in Table 4. A total of 37 mutated genes was found, among which only 11 (*ADAMTS10*, *CARD6*, *NLRP2*, *OBSCN*, *SECISBP2L*, *USP40*, *AKAP9*, *ARID1A*, *BAP1*, *KDM5C*, and *SETD2*) showed correlations with various clinicopathological findings. The 11 genes included six of the 123 survival-specific genes and five of the top 21 genes in TCGA-KIRC. Furthermore, we examined the mutation frequencies of the 11 genes listed in Table 4 in the normal population using Koreanchip, which gives the genomic sequencing reads of 210,000 normal Koreans [21,22]. As a result of our population-based analysis, mutations in 11 genes found in ccRCC patients were not found in Koreans without ccRCC.

A gene associated with OS, *CARD6* (*p* = 0.04, *r* = −0.441), was the only one identified among 123 survival-specific mutant genes. The Pearson’s correlation test revealed that the presence of mutations in *CARD6* was inversely related to the period of OS (Appendix A). Patient P10, with *CARD6* mutations, had an OS of 14 months, which was the shortest of the 22 patients. The gene found to be associated with DFS (*p* = 0.029, *r* = −0.465) was *BAP1*. The Pearson’s correlation test showed that the presence of the *BAP1* mutation resulted in shorter DFS (Appendix A). The presence of mutation in *SECISBP2L* showed an association with mortality (*p* = 0.03, *r* = 0.463) (Appendix A). The gene *SETD2* showed an association with nuclear grade (*p* = 0.035, *r* = 0.451); the higher the grade, the more frequent the mutation (Appendix A). Genes associated with the sarcomatoid component were *NLRP2* (*p* = 0.00038, *r* = 0.690), *OBSCN* (*p* = 0.026, *r* = 0.474), *USP40* (*p* = 0.00038, *r* = 0.690), and *AKAP9* (*p* = 0.00038, *r* = 0.690) (Appendix A). In the presence of a sarcomatoid component in histological findings, mutations in one or more of these four genes increased. Patients with mutations in *NLRP2*, *USP40*, or *AKAP9* had a strong association with sarcomatoid components, and patients with mutations in *OBSCN* showed a moderate correlation with sarcomatoid components. The mutated genes associated with lymph node metastasis were *SECISBP2L* (*p* = 2.20e-16, *r* = 1.000) and *KDM5C* (*p* = 0.00038, *r* = 0.690); both of these showed a very strong association (Appendix A). The Pearson’s correlation test revealed that the presence of mutations in *SECISBP2L* was closely related to the spread of cancer cells to the lymph nodes and also showed a significant association with mortality (*p* = 0.03, *r* = 0.463). Mutations in *OBSCN* were moderately associated with sex (*p* = 0.03, *r* = 0.462) as both patients with *OBSCN* mutations were female (P6 and P12) (Appendix A). The genes significantly associated with tumor size were *ADAMTS10* (*p* = 0.004, *r* = 0.585) and *ARID1A* (*p* = 0.004, *r* = 0.585) (Appendix A). *ADAMTS10* and *ARID1A* mutations were observed simultaneously in the patient (P18) who had the largest tumor size (14.5 cm) among the 22 patients (size of tumors without mutation, mean: 6.2 cm, median: 5.5 cm).

We identified the types and frequencies of mutations in 11 clinicopathologically significant genes in 22 Korean ccRCC patients and compared them with those reported by TCGA-KIRC and RECA-EU (Table 5). In this study, a mutation in *CARD6* (missense_variant; c.2674G>A, p.Gly892Arg) was found in 1 of 22 patients (4.5%, P10). This was identified as the only survival-specific gene. *OBSCN* mutations were observed in 2 of 22 patients (9.1%), both of whom were female. Mutations in both *ADAMTS10* and *ARID1A* genes were found in patient P18. The tumor in this patient was 14.5 cm, the largest among the 22 patients. Therefore, if there is a mutation in these two genes, the size of the tumor is expected to be large. In addition, various types of *SETD2* mutations (missense, frameshift, stop gained, splice variant, intron variant, and in-frame deletion) were observed, and the mutation frequency was 50%, more than double that reported in TCGA (21%) or RECA-EU (20%).

Table 6 compares the associations of survival rates obtained from TCGA-KIRC (Firehose Legacy and PanCancer Atlas) and RECA-EU with the 11 clinicopathologically significant genes. Analysis of the TCGA-KIRC cohort confirmed that all six survival-specific genes were significantly associated with survival rate, OS, and/or DFS in ccRCC patients. However, in the RECA-EU data, only four genes (*ADAMTS10*, *CARD6*, *OBSCN*, and *USP40*) showed a significant association with survival rate. In addition, the top-ranked mutated genes showed little correlation with survival rate in the TCGA data, and only *SETD2* showed a significant association with DFS. However, in the RECA-EU data, the top-ranked mutant genes showed a correlation with survival rate, OS, and/or DFS in all five genes.

The six survival-specific mutated genes (*ADAMTS10*, *CARD6*, *NLRP2*, *OBSCN*, *SECISBP2L*, and *USP40*) identified in TCGA-KIRC (Firehose Legacy and PanCancer Atlas) are shown in the survival graphs (Figure 2). Individual survival graphs showed that patients with mutations in each of the six genes had lower survival rates than patients without mutations for OS (*p* = 0.0186, *p* = 0.0003, *p* = 0.0029, *p* = 0.0096, *p* = 0.0005, and *p* = 0.378) and DFS (*p* = 0.0004, *p* = 0.074, *p* = 3.09e-12, *p* = 0.0048, *p* = 0.0489, and *p* = 0.0003). In addition, Figure 3 shows the survival graphs according to the presence or absence of mutations in all six survival-specific genes in TCGA-KIRC. Patients with mutations in at least one of the six survival-specific genes had lower rates for both OS (Firehose Legacy; *p* = 0.004, PanCancer Atlas; *p* = 3.166e-6) and DFS (Firehose Legacy; *p* = 5.65e-9, PanCancer Atlas; *p* = 0.0025) than those without mutations.

The hazard ratios (HR) and *p*-values from the multivariate Cox regression analysis of six survival-specific genes and two clinical factors, T stage and metastasis, are shown in Table 7. Mutations in *NLRP2* and *OBSCN* were significant in OS (HR = 5.72, *p* = 0.01 and HR = 7.5, *p* = 0.01), whereas *NLRP2* and *USP40* genes were significant in DFS (HR = 9.62 and *p* = <0.005 and HR = 5.29, *p* = 0.01). Additionally, it was noticed that mutations in *NLRP2* and *OBSCN* showed a statistically significant co-occurrence tendency (*p* = 0.042; Log2 Odds Ratio >3) from the Mutual Exclusivity analysis by cBioPortal (Zhiping Gu (2016), “cBioPortal,” https://ncihub.org/resources/1624, accessed on 15 November 2021). We identified that *NLRP2* gene mutations are significant in both OS and DFS. In Figure 4, *NLRP2* and *OBSCN* gene mutations have a higher risk ratio in OS than the two clinical factors. Similarly, *NLRP2* and *USP40* mutations have a higher risk ratio in DFS than both clinical factors.

## 4. Discussion

Although various studies are being conducted on genes related to survival in RCC, there are few survival-specific genes that can be used as biomarkers in clinical practice. In this study, we identified mutations in 22 Korean ccRCC patients using an NGS gene panel, which includes a total of 123 survival-specific genes discovered through machine learning with TCGA-KIRC data. Moreover, the relationship between these mutated genes and clinicopathological findings was investigated. However, the results were different from our initial expectations. The gene panel used in this study was developed based on American data from TCGA, as there are still few studies of ccRCC based on Asian populations. Therefore, when this panel is tested in Korean or European cohorts, the different effects of genes on survival are thought to be due to genetic differences between races.

According to the TCGA report [7], the most frequent gene mutations in kidney cancer were *VHL*, *PBRM1*, *SETD2*, and *BAP1*. In this study of 22 Korean patients, the overall tendency of mutation frequencies was different from other reports. In particular, four genes, *VHL* (91%), *SETD2* (50%), *TSHZ3* (14%), and *SPEN* (18%), that are among the 16 top-ranked mutated genes were observed with higher frequencies in Koreans than in other populations (e.g., USA, Europe, and the rest of Asia). However, it is not appropriate to interpret the observed differences in mutation frequency between regions of these genes only as regional differences. Since sample sizes vary widely between regions, the possibility that sample sizes are also relevant should be considered. Since the number of Korean patients (22 patients) participating in this study was small, future cohort expansion is necessary to derive a more definitive interpretation.

The aim of this study was to determine the presence or absence of mutations in the 123 survival-specific genes in 22 Korean ccRCC patients and to determine whether these gene mutations were related to survival. From 11 genes of clinical importance, six (*ADAMTS10*, *CARD6*, *NLRP2*, *OBSCN*, *SECISBP2L*, and *USP40*) were derived from 123 survival-specific genes, and the remaining five (*AKAP9*, *ARID1A*, *BAP1*, *KDM5C*, and *SETD2*) were derived from the 16 top-ranked mutated genes in TCGA_KIRC. Our study showed that the mutations in the 11 genes were correlated with factors of OS, DFS, mortality, nuclear grade, sarcomatoid component, N-stage, sex, and tumor size.

In this NGS study of 22 Korean ccRCC patients, only the *CARD6* mutation was found to correlate with OS (*p* = 0.04, *r* = −0.441) among 123 survival-specific genes. When we validated it with the publicly available data, there was a significant association between the *CARD6* gene mutation and OS in both TCGA (*p* = 0.0003) and RECA-EU (*p* = 0.0005) dataset. Therefore, in all three datasets, *CARD6* was the only gene that showed a significant association with survival (OS) in Korean, TCGA-KIRC (American), and RECA-EU data (European).

However, to date, no association between *CARD6* and survival has been reported in ccRCC or other cancers. *CARD6* (Caspase recruitment domain family member 6) is a CARD-containing protein and plays pivotal roles in signal transduction, leading to apoptosis and NF-κB activation and inflammation [23]. *CARD6* was reported to be involved in the activation of NF-κB signaling in gastrointestinal cancer [24]. In that study, neoexpression of *CARD6* was related to activation of the NF-κB pathway and played a role in the development of esophageal, gastric, and colorectal tumors.

In addition to *CARD6*, we found 10 other genes (*ADAMTS10*, *NLRP2*, *OBSCN*, *SECISBP2L*, *USP40*, *AKAP9*, *ARID1A*, *BAP1*, *KDM5C*, and *SETD2*) that revealed a correlation with OS and/or DFS. The *BAP1* (BRCA1 associated protein-1) protein functions as a deubiquinating enzyme that regulates multiple cellular pathways related to tumorigenesis [25]. Our finding that mutant *BAP1* was associated with DFS (*p* = 0.029, *r* = −0.465) was reported in other studies. In a comprehensive analysis of 445 ccRCC cases from TCGA, *BAP1* mutant ccRCC patients had worse OS (*p* = 0.035) and DFS (*p* = 0.036) than *BAP1* wild-type patients [26]. *BAP1* mutations were associated with worse cancer-specific survival (CSS) in both cohorts of the Memorial Sloan-Kettering Cancer Center (*p* = 0.002; HR 7.71; 95% confidence interval (CI) 2.08–28.6) and TCGA (*p* = 0.002; HR 2.21; 95% CI 1.35–3.63) [8,27]. Kapur et al. reported that tumors with negative *BAP1* expression (most were *BAP1* mutants) were related to shorter OS (*p* = 0.001) [28], and 90% of *BAP1*-mutant tumors were related to a high (grades 3–4) Fuhrman grade [25]. In addition, as in the 145 patients with primary ccRCC from the University of Texas Southwestern Medical Center, patients with *BAP1*-mutated tumors had a significantly higher probability of death (HR 2.8, 1.4–5.9; *p* = 0.004) and were associated with a higher grade (*p* = 0.095), necrosis (*p* = 0.038), and advanced pathologic tumor stage and clinical stage (*p* = 0.011 and *p* = 0.003, respectively), indicating that *BAP1*-mutant tumors were associated with poor outcome [9]. In particular, a multivariate Cox proportional hazards model suggested that patients with *NLRP2*, *USP40*, and *OBSCN* mutations have a higher mortality rate than other factors.

*SECISBP2L* (SECIS binding protein 2-like) is a paralogue of SECIS binding protein 2 and is known to play a role in selenoprotein expression [29]. The association of the *SECISBP2L* mutation found in our previous study using the TCGA_KIRC cohort with OS (*p* = 0.0005) and DFS (*p* = 0.0489) was not identified in 22 Korean patients. To date, there have been no reports of an association between *SECISBP2L* mutation and survival or mortality.

*SETD2*, a histone methyltransferase, plays an important role in the epigenetic control of gene expression [27]. *SETD2* mutations, observed to be associated with nuclear grade (*p* = 0.035, *r* = 0.451) in this study, were not reported in other studies of ccRCC. It was reported that mutations in *SETD2* occurred in 3–12% of ccRCC cases and were associated with poor clinical outcome [30]. Decreased *SETD2* expression predicted unfavorable prognosis (larger tumor size and advanced pT stage) in patients with ccRCC [30]. *SETD2* was associated with worse CSS in a TCGA cohort of 421 patients with primary ccRCC (*p* = 0.036; HR 1.68; 95% CI 1.04–2.73) [8]. Hakimi et al. reported that tumors with mutations in *BAP1*, *SETD2*, or *KDM5C* were more likely to present with stage III disease or higher (*p* = 0.001) [27]. We also found that *SETD2* correlated with DFS in TCGA (*p* = 0.0201) and RECA-EU (*p* = 0.0002). Similarly, a recent Chinese ccRCC study reported a correlation between a *SETD2* mutation and a shorter DFS (*p* = 0.065) [6]. They performed deep sequencing, targeting 556 oncogenes in 105 tumor tissues to detect somatic mutations and clinicopathological effects in Chinese patients with ccRCC. Eight genes (*BAP1*, *SETD2*, *PTEN*, *ERBB2*, *TP53*, *CDK8*, *TSC1*, and *SPEN*) were associated with poor prognosis [27].

Our study showed a significant correlation of mutations in *NLRP2*, *USP40*, *AKAP9* (*r* = 0.690, *p* = 0.00038), and *OBSCN* (*r* = 0.474, *p* = 0.026) with a sarcomatoid component. To our knowledge, there were no reports of associations between mutations in *NLRP2*, *OBSCN*, *USP40*, and *AKAP9* genes and the sarcomatoid component. However, Liu et al. reported that the combined score according to *SETD2*/H3K36me3 expression was an independent prognostic factor for OS and DFS, which was associated with tumor size (*p* = 0.003), pT stage (*p* = 0.043), and sarcomatoid component (*p* = 0.004) [30].

The nod-like receptor pyrin domain-containing proteins (*NLRPs*) are expressed by resident renal cells. Overexpression of the NLRP family member *NLRP2* was found to contribute to the progression of renal failure by creating a vicious inflammatory cycle and decreasing the apoptotic cell rate [31]. Although we found that *NLRP2* mutations are correlated with OS (*p* = 0.0029) and DFS (*p* = 3.09e-12) in the TCG-KIRC data, only a few studies have reported similar data, but not in RCC. A lung adenocarcinoma study has reported that low expression of *NLRP2* is correlated with poor survival rates in lung adenocarcinoma (*p* = 0.014) [32]. High expression of *NLRP2* was associated with high risk in head and neck squamous cell carcinoma [33].

The *OBSCN* (Obscurin, cytoskeletal calmodulin and titin-interacting RhoGEF) gene encodes obscurin, which belongs to a family of giant cytoskeletal proteins [34]. *OBSCN* is regarded as a tumor suppressor for its ability to influence cellular integration and activate cancer initiation [35]. We discovered that *OBSCN* mutations were correlated with OS (*p* = 0.0096) and DFS (*p* = 0.0048) in the TCGA-KIRC cohort, and with DFS (*p* = 3.28e-6) in the RECA-EU cohort, although there was no correlation of *OBSCN* mutations with survival in our 22 Korean ccRCC patients. Even though current studies on *OBSCN* mutations are minimal, the association between *OBSCN* expression and clinical factors has been reported in other studies. A previous pRCC study reported that higher *OBSCN* expression was associated with poor survival outcomes (*p* = 0.021) [35]. A pancreatic adenosquamous carcinoma study reported that somatic mutations in *OBSCN* had a negative correlation with lymphatic metastasis in pancreatic adenosquamous carcinoma (*p* = 0.0339) [34]. Also, in our results, the *OBSCN* mutation was associated with female sex (*p* = 0.03, *r* = 0.462), but similar results have not been reported.

The *USP40* (Ubiquitin-specific peptidase 40) protein is a deubiquitinase, regulating the ubiquitination and degradation of CFLARL, which plays an important role in extrinsic ligand-induced apoptosis [36]. We found that the *USP40* mutation was correlated with DFS (*p* = 0.0003 and *p* = 0.0269) in the TCGA_KIRC cohort (Firehose Legacy and PanCancer Atlas) and with OS (*p* = 0.042) in the RECA-EU cohort. However, this was not validated in our 22 ccRCC patients.

The A-kinase anchor proteins (*AKAPs*) have the common function of binding to the regulatory subunit of protein kinase A and confining the holoenzyme to discrete locations within the cell. *AKAP9* has been reported to be involved in the development or metastasis of several cancers [37]. We discovered that the *AKAP9* mutation was correlated with OS (*p* = 0.047) in the RECA-EU cohort, but not in the TCGA_KIRC cohort or our 22 Korean ccRCC patients.

In this study, we identified *SECISBP2L* (*p* = 2.20e-16, *r* = 1.000) and *KDM5C* (*p* = 0.00038, *r* = 0.690) as genes associated with lymph node metastasis. A correlation between mutations of *SECISBP2L* or *KDM5C* and N-stage has not yet been reported in RCC. However, a recent colon cancer study stated that *SECISBP2L* showed high expression in the LN (+) group rather than the LN (−) group, and the expression level was significantly correlated with the survival rate [38]. *KDM5C* (Lysine demethylase 5C) is a JmjC domain-containing protein that removes methyl residues from methylated lysine 4 on histone H3 lysine 4. *KDM5C* has been proposed as an oncogene in many types of tumors [39]. We found that the *KDM5C* mutation was correlated with OS (*p* = 0.022) in a RECA-EU cohort, but not in a TCGA-KIRC cohort or our 22 Korean ccRCC patients. In a ccRCC study, Hakimi et al. reported a significant connection (*p* = 0.001) between tumors with any of the three mutations of *KDM5C*, *BAP1*, or *SETD2* and advanced stages [27]. It was reported that mutations in *KDM5C* and *BAP1* were significantly associated with renal vein invasion (*p* = 0.022 and *p* = 0.046, respectively) [40].

In our study, the two genes *ADAMTS10* and *ARID1A* (both *p* = 0.004, *r* = 0.585) were associated with tumor size. The finding that *ADAMTS10* mutations were associated with large tumor size has not been reported in other studies. However, there was a significant correlation between *ARID1A* mRNA expression and size (*p* = 0.03), grade (*p* = 0.03), and stage (*p* = 0.03) of tumor in a ccRCC study [41]. Liu et al. also reported that patients with low expression of *SETD2* or H3K36me3 were prone to a large tumor size and advanced pT stage [30]. The A disintegrin and metalloproteinase with thrombospondin motif (*ADAMTS*) family of genes are a group of proteases found in both mammals and invertebrates [42] and reported to be involved in the occurrence and development of different cancers [43]. Although we discovered that *ADAMTS10* mutations were correlated with OS (*p* = 0.0186) and DFS (*p* = 0.0004) in the TCGA-KIRC cohort, and with DFS (*p* = 0.0011) in the RECA-EU cohort, there was no correlation in the 22 Korean ccRCC patients. However, Liang et al. reported that the expression level of *ADAMTS10* affects OS (HR = 1.45, *p* < 0.026) and recurrence (HR = 2.22, *p* = 0.020) in patients with gastric cancer. The survival of patients with a high expression of *ADAMTS10* was significantly lower than that of patients with a low expression [43].

The *ARID1A* (AT-rich interaction domain 1A), coding for the BAF250a subunit of BAF, is a tumor suppressor gene often mutated in RCC [41] and other types of carcinoma [44]. We found an association between *ARID1A* mutations and larger tumor sizes in this study (*p* = 0.004, *r* = 0.585). According to a report on the association of *ARID1A* with tumor size in ccRCC [41], decreased *ARID1A* expression was associated with larger tumors (≥7 cm), high grades III and IV, and high stage III and IV (all: *p* = 0.03). Additionally, we discovered that *ARID1A* mutation was correlated with OS (*p* = 0.0204) and DFS (*p* = 1.24e-5) in the RECA-EU cohort, but not in the TCGA-KIRC cohort or the 22 Korean ccRCC patients. A significantly lower expression of *ARID1A* than matched normal kidney cortex was reported in 67% of ccRCC (53 of 79), and a Kaplan–Meier analysis demonstrated significantly shorter OS among patients with *ARID1A*-negative tumors (x^2^ = 9.14; *p* = 0.003) [41]. The study of *ARID1A* in 290 cases of ccRCC in Korea showed that lower *ARID1A* expression was associated with higher nuclear grade (*p* < 0.001), higher pTNM stage (*p* = 0.013), and shorter CSS (*p* = 0.001) and progression-free survival (*p* < 0.001) [45].

Recently, radiogenomic studies of ccRCC showed associations between CT imaging features and mutations [40]. Mutations of *VHL* were significantly associated with well-defined tumor margins (*p* = 0.013), nodular tumor enhancement (*p* = 0.021), and gross appearance of intratumoral vascularity (*p* = 0.018). The solid type of ccRCC differed in genotype significantly from the multicystic type. While mutations in *BAP1*, *SETD2*, and *KDM5C* were absent in the multicystic type, mutations of *VHL* (*p* = 0.016) and *PBRM1* (*p* = 0.017) were significantly more common in the solid type [40].

This study confirmed that genetic mutations in particular genes are related to survival and clinicopathological findings in diverse forms. The NGS technique and population-based analysis revealed that genetic differences, frequencies, and clinical differences in ccRCC exist between cohorts in East Asia, North America, and Europe. The limitation of the present study is that the panel was developed with genes derived from the TCGA-KIRC database due to a lack of studies on Asians with ccRCC. This accounts for the different results in survival-specific genes between the Korean and the Western cohorts. Another limitation is the size of the Korean samples used in this study. Further studies in large patient cohorts are needed to verify the clinicopathological significance of these novel mutations and to develop targeted therapeutics using these genetic markers. The authors expect that, based on the results of this study, a gene panel that predicts the prognosis of ccRCC patients will be developed and utilized by clinicians, including pathologists, urologists, oncologists, and radiation oncologists. However, many challenges remain to provide a reliable prognostic strategy for ccRCC patients.

In conclusion, we discovered and validated 11 survival-specific genes in ccRCC patients by using data from the TCGA-KIRC, RECA-EU, and Korean patients. To our knowledge, we are the first to find a correlation between *CARD6* mutations and OS in patients with ccRCC. The 11 genes, including *CARD6*, *NLRP2*, *OBSCN*, and *USP40*, might be useful diagnostic, prognostic, and therapeutic markers in ccRCC.

## Figures and Tables

**Figure 1 jpm-12-00113-f001:**
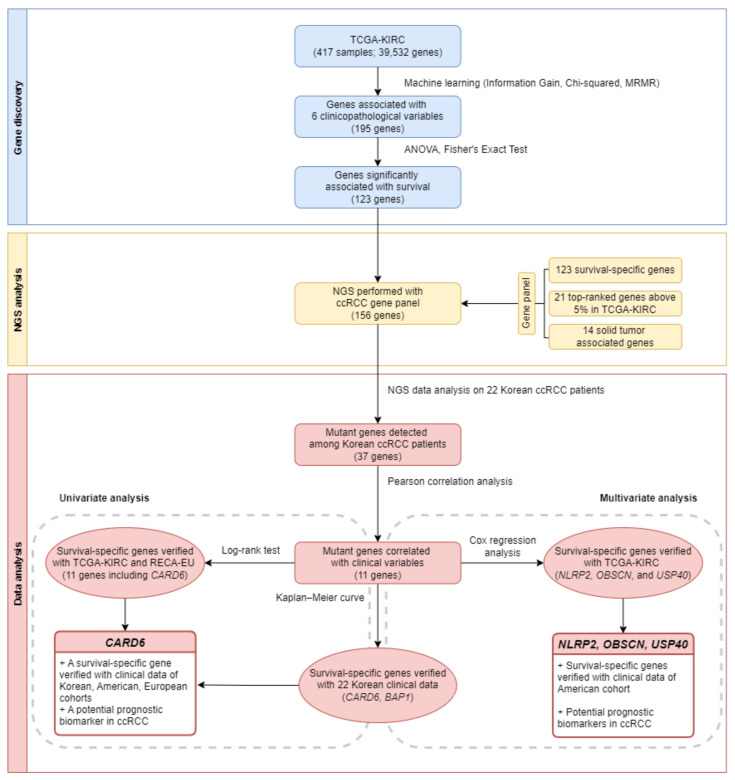
Workflow of the study. NGS, Next Generation Sequencing; TCGA-KIRC, The Cancer Genome Atlas Kidney Renal Clear Cell Carcinoma; MRMR, Minimum Redundancy and Maximum Relevance; ANOVA, Analysis of variance; ccRCC, clear cell renal cell carcinoma; RECA-EU, Renal Cell Cancer-European Union.

**Figure 2 jpm-12-00113-f002:**
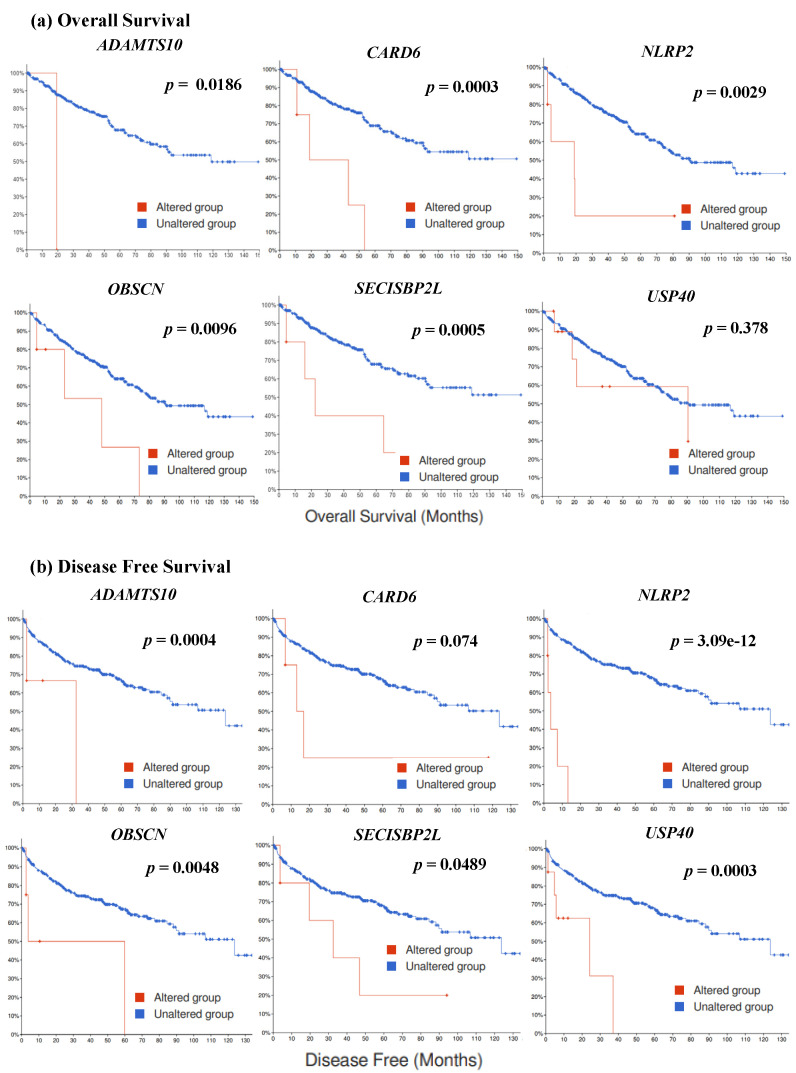
Survival analysis of TCGA_KIRC data for six survival-specific mutated genes. (**a**) Overall survival graphs of *ADAMTS10*, *CARD6*, and *SECISBP2L* were obtained from the PanCancer Atlas; overall survival graphs of *NLRP2*, *OBSCN*, and *USP40* were obtained from TCGA, Firehose Legacy. (**b**) Disease-free survival graphs of *ADAMTS10*, *CARD6*, *NLRP2*, *OBSCN*, *SECISBP2L*, and *USP40* were based on data from TCGA, Firehose Legacy.

**Figure 3 jpm-12-00113-f003:**
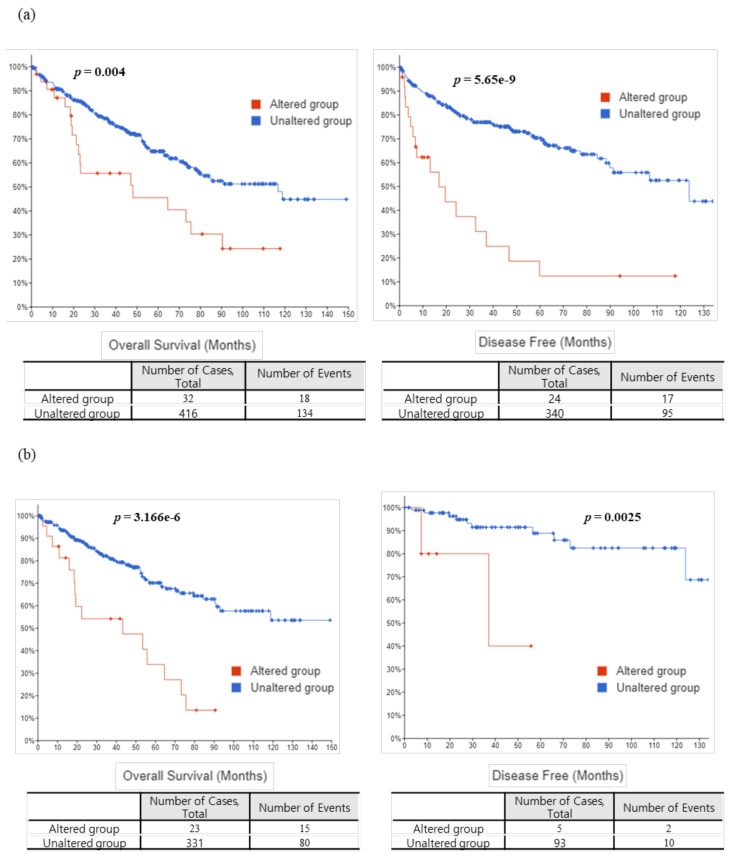
Survival analysis of the TCGA-KIRC data for all six survival genes (*ADAMTS10*, *CARD6*, *NLRP2*, *OBSCN*, *SECISBP2L*, and *USP40*). Overall survival (left) and disease-free survival (right) graphs based on data from (**a**) the Firehose Legacy and (**b**) the PanCancer Atlas.

**Figure 4 jpm-12-00113-f004:**
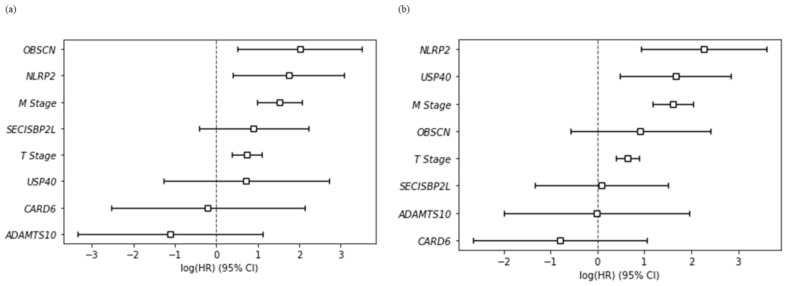
Multivariate forest plots on survival in TCGA_KIRC (Firehose Legacy) data by the multivariate Cox regression analysis: (**a**) overall survival and (**b**) disease-free survival. CI, confidence interval; log(HR), hazard ratio.

**Table 1 jpm-12-00113-t001:** Clinicopathological findings of 22 Korean patients with clear cell renal cell carcinoma.

	**P01**	**P02**	**P03**	**P04**	**P05**	**P06**	**P07**	**P08**	**P09**	**P10**	**P11**
Age	68	60	65	72	69	33	42	40	52	57	68
Sex (F/M)	F	F	M	F	M	F	F	M	M	M	M
Tumor Size (cm)	2.5	2.5	3.9	11	5.5	4.3	6	5.3	8.8	5.2	10
Nuclear Grade	II	II	IV	IV	III	II	II	III	III	III	IV
Sarcomatoid component	0	0	0	0	1	0	0	0	0	0	0
TNM T-Stage	T1a	T1a	T3a	T2b	T1b	T1b	T1b	T1b	T2a	T3a	T3a
TNM N-Stage	0	0	0	0	0	0	0	0	0	0	1
TNM M-Stage	0	0	0	0	1	0	0	0	1	0	1
Recurrence	No	No	No	No	Lung, Bone	No	No	No	Liver, Brain, Lung	No	Lung, Liver, Abdomen, Retroperitoneum
Overall Survival (Month)	36	35	35	35	32	32	30	28	12	14	42
Disease Free Survival (Month)	36	35	35	35	10	32	30	28	10	14	10
Death	N	N	N	N	N	N	N	N	Y	N	Y
Response to LRN *	NED	NED	NED	NED	Fail	NED	NED	NED	Fail	NED	Fail
	**P12**	**P13**	**P14**	**P15**	**P16**	**P17**	**P18**	**P19**	**P20**	**P21**	**P22**
Age	67	62	51	40	82	82	65	47	45	82	61
Sex (F/M)	F	M	M	M	F	M	M	M	M	M	M
Tumor Size (cm)	6	10	5	4.2	5.5	4.7	14.5	10.3	7.5	7.2	5.4
Nuclear Grade	IV	III	III	III	II	III	II	III	III	III	IV
Sarcomatoid component	1	0	0	0	0	0	0	0	0	0	0
TNM T-Stage	T1b	T2a	T3a	T3a	T3a	T3a	T3b	T3a	T2a	T2a	T3b
TNM N-Stage	0	0	0	0	0	0	0	0	0	0	0
TNM M-Stage	1	1	1	0	1	0	0	1	0	1	1
Recurrence	Lung, Bone	Lung	Lung, Brain	No	Lung	No	No	Lung	No	Lung	Lung, Bone, Jejunum, Peritoneum
Overall Survival (Month)	42	38	37	36	14	38	48	50	45	48	34
Disease Free Survival (Month)	0	1	33	36	1	38	48	2	45	24	1
Death	N	N	N	N	Y	N	N	N	N	N	Y
Response to LRN *	Fail	Fail	Fail	NED	Fail	NED	NED	Fail	NED	Fail	Fail

LRN *: Laparoscopic Radical Nephrectomy, NED: No evidence of disease. P, patient; F/M, female/male; TNM, Tumor, node, metastasis; N, No; Y, Yes.

**Table 2 jpm-12-00113-t002:** Comparison of top-ranked mutated genes in clear cell renal cell carcinoma in 22 Korean patients and patients from other regions.

	Mutation Frequency (%)					
Gene	Korean	TCGA	RECA-EU	Tokyo	Taiwan	
*VHL*	91	54	61	41	50	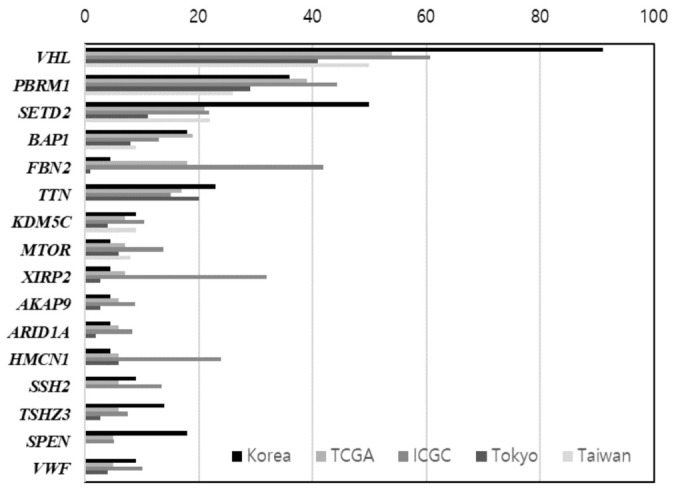
*PBRM1*	36	39	44	29	26	
*SETD2*	50	21	22	11	22	
*BAP1*	18	19	13	8	9	
*FBN2*	4.5	18	42	0.9	NA	
*TTN*	23	17	15	20	NA	
*KDM5C*	9	7	10	4	9	
*MTOR*	4.5	7	14	6	8	
*XIRP2*	4.5	7	32	2.8	NA	
*AKAP9*	4.5	6	9	2.8	NA	
*ARID1A*	4.5	6	8	1.9	NA	
*HMCN1*	4.5	6	24	6	NA	
*SSH2*	9	6	14	0	NA	
*TSHZ3*	14	6	8	2.8	NA	
*SPEN*	18	5	5	0	NA	
*VWF*	9	5	10	4	NA	

TCGA, The Cancer Genome Atlas; RECA-EU, Renal Cell Cancer-European Union; NA, not available; ICGC, International Cancer Genome Consortium.

**Table 3 jpm-12-00113-t003:** Types of mutations in *VHL* genes found in 22 Korean patients with clear cell renal cell carcinomas.

HGVS.c (cDNA)	HGVS.p (Protein)	HGVS.p (Single)	Variant Type	ClinVar	COSMIC ID
c.337C>T	p.Arg113	R113	Stop gained	Pathogenic	COSM30228
c.353T>C	p.Leu118Pro	L118P	Missense variant Intron variant	Pathogenic	COSM14312
c.174_208delGCCGC GGCCCGTG CTGCG CTCGGTGAACTCG CGCG	p.Pro59fs	P59fs	Frameshift variant	Unknown	-
c.463+1G>A	-	-	Intron variant	Pathogenic	COSM51391
c.263G>A	p.Trp88	W88	Stop gained	Pathogenic	COSM18070
c.220_231dupGTCAT CTTCTGC	p.Val74_Cys77dup	V74_C77dup	Conservative inframe insertion	Unknown	-
c.257C>T	p.Pro86Leu	P86L	Missense variant	Pathogenic	COSM18028
c.473T>A	p.Leu158Gln	L158Q	Missense variant	Likely pathogenic	COSM14368
c.227_229delTCT	p.Phe76del	F76del	Disruptive inframe deletion	Unknown	COSM53186
c.430G>T	p.Gly144	G144	Stop gained Intron variant	Pathogenic	COSM25682
c.449delA	p.Asn150fs	N150fs	Frameshift variant Intron variant	Unknown	COSM17843
c.332G>T	p.Ser111Gly	S111l	Missense variant	Uncertain significance	COSM36341
c.280delG	p.Glu94fs	E94fs	Frameshift variant	Unknown	-
c.281A>T	p.Glu94Val	E94V	Missense variant	Unknown	-
c.331A>G	p.Ser111Gly	S111G	Missense variant	Pathogenic	COSM18353
c.337C>T	p.Arg113	R113	Stop gained	Pathogenic	COSM30228
c.266T>A	p.Leu89His	L89H	Missense variant	Likely pathogenic	COSM14305
c.523dupT	p.Tyr175fs	Y175fs	Frameshift variant	Unknown	COSM253386
c.203C>A	p.Ser68	S68	Stop gained	Pathogenic	COSM14372
c.362A>G	p.Asp121Gly	D121G	Missense variant intron variant	Pathogenic	COSM18009

**Table 4 jpm-12-00113-t004:** Eleven clinicopathologically significant mutated genes found in 22 Korean patients with clear cell renal cell carcinoma.

Clinical Variable	Result	Survival-Specific Mutated Genes of ccRCC						Top Ranked Mutated Genes of TCGA_KIRC				
		*ADAMTS10*	*CARD6*	*NLRP2*	*OBSCN*	*SECISBP2L*	*USP40*	*AKAP9*	*ARID1A*	*BAP1*	*KDM5C*	*SETD2*
Overall Survival	*r*	0.287	**−0.441** *	−0.055	0.119	0.159	–0.055	−0.055	0.287	−0.235	−0.204	0.377
	*p*-value	0.195	**0.04**	0.806	0.597	0.481	0.806	0.806	0.195	0.291	0.361	0.084
Disease Free Survival	*r*	0.350	−0.124	−0.180	−0.234	−0.180	−0.180	−0.180	−0.350	**−0.465** *	−0.221	−0.115
	*p*-value	0.110	0.581	0.422	0.295	0.422	0.422	0.422	0.110	**0.029**	0.323	0.610
Death	*r*	−0.087	−0.087	−0.087	−0.126	**0.463** *	−0.087	−0.087	−0.087	0.389	−0.126	0.024
	*p*-value	0.701	0.701	0.701	0.578	**0.03**	0.701	0.701	0.701	0.073	0.577	0.916
Nuclear Grade	*r*	−0.299	0.019	0.019	0.034	0.318	0.019	0.019	−0.299	0.192	0.244	**0.451** *
	*p*-value	0.176	0.934	0.934	0.881	0.149	0.934	0.934	0.176	0.392	0.274	**0.035**
Sarcomatoid component	*r*	−0.069	−0.069	**0.690** **	**0.474** *	−0.069	**0.690** **	**0.690** **	−0.069	0.261	−0.1	0.314
	*p*-value	0.76	0.76	**0.00038**	**0.026**	0.76	**0.00038**	**0.00038**	0.76	0.241	0.658	0.155
N-stage	*r*	−0.048	−0.048	−0.048	−0.069	**1.000** **	−0.048	−0.048	−0.048	−0.103	**0.690** **	0.217
	*p*-value	0.833	0.833	0.833	0.761	**2.20e-16**	0.833	0.833	0.833	0.649	**0.00038**	0.333
Sex	*r*	−0.149	−0.149	−0.149	**0.462** *	−0.149	−0.149	−0.149	−0.149	0.184	-0.216	−0.379
	*p*-value	0.508	0.508	0.508	**0.03**	0.508	0.508	0.508	0.508	0.412	0.334	0.082
Tumor size	*r*	**0.585** **	−0.104	−0.082	−0.119	0.252	−0.082	−0.082	**0.585** **	−0.169	0.107	0.123
	*p*-value	**0.004**	0.645	0.717	0.599	0.259	0.717	0.717	**0.004**	0.453	0.636	0.585

* is used for *p*-values < 0.05. ** are used for *p*-values < 0.001. Statistical significant values are bolded. The *p*-values are from Pearson’s correlation analysis.

**Table 5 jpm-12-00113-t005:** Types and frequencies of 11 clinically significant mutated genes found in 22 Korean patients with clear cell renal cell carcinoma.

Genes	Mutant Type	Mutation	HGVS.p	Mutation Frequency %	Patient ID	TCGA-KIRC (Firehose) Mutation Frequency %	RECA-EU Mutation Frequency %
		HGVS.c					
*ADAMTS10*	missense_variant	c.2303G>A	p.Arg768His	4.5	P18	0.9	2.1
*CARD6*	missense_variant	c.2674G>A	p.Gly892Arg	4.5	P10	1.1	0.7
*NLRP2*	missense_variant	c.2233C>T	p.Arg745Trp	4.5	P05	1.1	4.7
*OBSCN*	missense_variant	c.18052C>T	p.Arg6018Cys	9.1	P06	1.1	6
		c.3529C>G	p.Gln1177Glu		P12		
		c.12072_12073delTCinsCT	p.Arg4025Cys				
*SECISBP2L*	missense_variant	c.1930A>G	p.Met644Val	4.5	P11	1.8	3.5
*USP40*	frameshift_variant	c.3477dupA	p.Gln1160fs	4.5	P05	2.2	4.5
*AKAP9*	missense_variant	c.89A>T	p.Gln30Leu	4.5	P05	6	7
*ARID1A*	missense_variant	c.6200T>G	p.Ile2067Ser	4.5	P18	6	7
*BAP1*	missense_variant and splice_region_variant	c.437G>C	p.Arg146Thr	18.1	P08	19	13
	frameshift_variant	c.878dupC	p.Leu294fs		P12		
		c.1636_1642delTACAACC	p.Tyr546fs		P16		
	splice_acceptor_variant and splice_region_variant	c.581-2A>G	Unknown		P22		
*KDM5C*	frameshift_variant	c.3460delG	p.Glu1154fs	9.1	P10	7	10
	splice_donor_variant&intron_variant	c.531+2T>A	Unknown		P11		
*SETD2*	missense_variant	c.2357_2358delGCinsTT	p.Cys786Phe	50	P08	21	20
		c.4885C>G	p.His1629Asp		P18		
		c.373T>G	p.Ser125Ala		P21		
		c.577C>T	p.Pro193Ser		P22		
	frameshift_variant	c.7537_7546dupACTCACGGTG	p.Val2516fs		P11		
		c.572delC	p.Pro191fs		P22		
	stop_gained	c.6520C>T	p.Gln2174		P05		
		c.4486C>T	p.Arg1496		P21		
	stop_gained and splice_region_variant	c.5013T>G	p.Tyr1671		P12		
	splice_donor_variant and intron_variant	c.4715+1G>A	Unknown		P03		
	conservative_inframe_deletion	C.625_681delACAGAGCCAGTGGCCTTGCCACATACACCAATAACAGTTCTAATGGCAGCACCAGTA	p.Thr209_Val227del		P20		

TCGA-KIRC, The Cancer Genome Atlas Kidney Renal Clear Cell Carcinoma; RECA-EU, Renal Cell Cancer-European Union; P, patient.

**Table 6 jpm-12-00113-t006:** Comparison of survival analysis of TCGA-KIRC and RECA-EU data for 11 clinicopathologically significant mutated genes found in 22 Korean patients with clear cell renal cell carcinoma.

Survival Analysis		Survival-Specific Mutated Genes of ccRCC						Top Ranked Mutated Genes of TCGA_KIRC				
		*ADAMTS10*	*CARD6*	*NLRP2*	*OBSCN*	*SECISBP2L*	*USP40*	*AKAP9*	*ARID1A*	*BAP1*	*KDM5C*	*SETD2*
TCGA_KIRC	Overall Survival											
	Firehose Legacy	0.209	0.15	**0.0029** *	**0.0096** *	0.112	0.378	0.284	0.906	0.577	0.102	0.821
	PanCancer Atlas	**0.0186** *	**0.0003** **	**0.0142** *	0.289	**0.0005** **	0.53	0.845	0.464	0.0754	0.0664	0.273
	Disease Free Survival											
	Firehose Legacy	**0.0004** *	0.074	**3.09e-12** **	**0.0048** *	**0.0489** *	**0.0003** **	0.8	0.384	0.285	0.0911	0.238
	PanCancer Atlas	NA	NA	**<10^−10^** **	0.695	NA	**0.0269** *	0.177	0.228	0.401	0.488	**0.0201** *
RECA-EU	Overall Survival	0.326	**0.0005** **	0.061	0.423	0.564	**0.042** *	**0.047** *	**0.0204** *	**0.0024** *	**0.022** *	0.056
	Disease Free Survival	**0.0011** *	1.0	0.65	**3.28e-06** **	0.669	0.669	0.263	**1.24e-05** **	0.421	0.491	**0.0002** **

* is used for *p*-values < 0.05. ** are used for *p*-values < 0.001. Statistical significant values are bolded. The *p*-values are from the Log rank test. ccRCC, clear cell renal cell carcinoma; TCGA_KIRC, The Cancer Genome Atlas Kidney Renal Clear Cell Carcinoma; RECA-EU, Renal Cell Cancer-European Union; NA, Not available.

**Table 7 jpm-12-00113-t007:** Multivariate Cox regression analysis on survival in six survival-specific genes from TCGA-KIRC (Firehose Legacy) data.

Multivariate Cox Regression						
	Overall Survival			Disease Free Survival		
Covariates	Hazard Ratio	Lower-Upper (95% CI)	*p*-Value	Hazard Ratio	Lower-Upper (95% CI)	*p*-Value
*ADAMTS10*	0.33	0.03–3.08	0.33	0.98	0.14–7.07	0.98
*CARD6*	0.82	0.08–8.44	0.86	0.45	0.07–2.85	0.39
*NLRP2*	5.72	1.48–22.1	**0.01 ***	9.62	2.51–36.92	**<0.005 ***
*OBSCN*	7.5	1.67–33.69	**0.01 ***	2.5	0.56–11.09	0.23
*SECISBP2L*	2.47	0.66–9.22	0.18	1.09	0.27–4.51	0.9
*USP40*	2.06	0.28–15.08	0.48	5.29	1.62–17.29	**0.01 ***
T Stage	2.09	1.46–3	**<0.005 ***	1.9	1.48–2.44	**<0.005 ***
Metastasis	4.62	2.68–7.95	**<0.005 ***	4.98	3.21–7.74	**<0.005 ***

* is used of *p*-values < 0.05. Statistical significant values are bolded. The *p*-values are from the Cox proportional hazard model. CI, confidence interval.

## Data Availability

This study was conducted with bio-resources from National Biobank of Korea, the Centers for Disease Control and Prevention, Republic of Korea (KBN-2019-019, approval date: 21 March 2019; Project Number, KC18SNSI0549).

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
