# Peer review of "Identification of Survival-Specific Genes in Clear Cell Renal Cell Carcinoma Using a Customized Next-Generation Sequencing Gene Panel"

_jpm, 2022, doi:10.3390/jpm12010113_

Round 1
Reviewer 1 Report
Clear cell renal cell carcinoma (ccRCC) is one of the most lethal cancer types globally, accounting for 70-75% of all renal cancers. Studies about ccRCC are chiefly associated with mutations which are carcinogenic. Meanwhile, studies are based on white Caucasian. Therefore, there are few studies about survival of patients and Asian. This article developed a customized next generation sequencing (NGS) gene panel with 156 genes based on TCGA, RECA, ucsc database and the data acquired from Korean, aims to find survival-specific genes, and the relationship between these genes and the clinicopathology.
The article’s thoughts is clear relatively, argument is rich relatively, logic is strong relatively, but still have some problems to deal with.
Q1:Line 209-212: Which list are the 16 genes from?
Q2:The comparison of the mutation frequencies of the four genes in various regions has some problems: whether the difference among regions is significant can not be judged according to the proportion value, it should be judged in accordance with sample size. Only 22 Korean samples involved in this article, at the same times, there are hundreds samples from TCGA, so the difference is often not significant.
Q3:How is pearson correlation applied to mutation and survival data in Table 5?
Q4:Table 6 have not explained one of the purpose of this article: At present, the studies on the foreign species of renal transparent cell cancer are in the majority, the survival-specific genes are derived from foreign species, which may account for why survival-specific genes are different from Korean species and foreign species.
Q5:In Figure 1, compared with survival analysis, this experiment should focus on multivariate ANOVA. Whether the mutations of 6 genes are linked or affect each other. Survival analysis can not exclude the interference of other genes’ mutations.
Q6:There is still no focused explanation in discussion for why survival-related genes derived from the TCGA database, rarely perform survival differences in the Korean data.
Author Response
Q 1: Line 209-212: Which list are the 16 genes from?
Response 1: The 16 genes are from the list of 21 top-ranked mutant genes above 5% in TCGA-KIRC (Supplementary Table S3).
Q 2: The comparison of the mutation frequencies of the four genes in various regions has some problems: whether the difference among regions is significant cannot be judged according to the proportion value, it should be judged in accordance with sample size. Only 22 Korean samples involved in this article, at the same times, there are hundreds samples from TCGA, so the difference is often not significant.
Response 2: We have added the following explanation to reflect your comments on sample size (lines 362-367).
“However, it is not appropriate to interpret the observed differences in mutation frequency between regions of these genes only as regional differences. Since sample sizes vary widely between regions, the possibility that sample sizes are also relevant should be considered. Since the number of Korean patients (22 patients) participating in this study was small, it is considered that future cohort expansion is necessary to derive a more definitive interpretation.”
Q 3: How is pearson correlation applied to mutation and survival data in Table 5?
Response 3: Table 5 only indicates variant details of the 11 genes (types and mutation frequencies in two different databases, etc.), hence, Pearson’s correlation analysis is not relevant to this table.
Instead, Table 4 shows Pearson’s correlation coefficients (R) and p-values between 8 clinical variables and the 11 genes. The method we used was mentioned in the manuscript (lines 262).
Or, if you referred to Table 6 that compares survival graphs of the 11 genes using TCGA-KIRC and RECA-EU dataset, we used log-rank test to derive their statistical significance. The method used in this case was also mentioned in the manuscript (lines 315).
Q 4: Table 6 have not explained one of the purpose of this article: At present, the studies on the foreign species of renal transparent cell cancer are in the majority, the survival-specific genes are derived from foreign species, which may account for why survival-specific genes are different from Korean species and foreign species.
Response 4: We have added the following explanation to reflect your comments (lines 353-356).
“The gene panel used in this study was developed based on American data from TCGA, as there are still few studies of ccRCC based on Asian populations. Therefore, when this panel is tested in Korean or European cohorts, the different effects of genes on survival are thought to be due to genetic differences between races.”
Q 5: In Figure 1, compared with survival analysis, this experiment should focus on multivariate ANOVA. Whether the mutations of 6 genes are linked or affect each other. Survival analysis cannot exclude the interference of other genes’ mutations.
Response 5: To simultaneously consider multiple factors for survival analysis, we’ve performed the multivariate Cox proportional hazard regression for multivariate analysis and the following outcome has been added to the manuscript.
(lines 340-345): Table 7 and Figure 4
(lines 200-204)
"When performing survival analyses using datasets from TCGA-KIRC and RECA-EU, the Kaplan-Meier method and the multivariate Cox proportional hazard model were used. The survival curves and hazard ratios over the given survival periods were estimated by utilizing Python library called lifelines (https://lifelines.readthedocs.io/en/latest/)."
(lines 330-339)
“The outcome from multivariate Cox regression analysis of six survival-specific genes and two clinical factors, T stage and metastasis, is shown in Table 7. Mutations in NLRP2 and OBSCN were significant in OS (HR = 5.72, P = 0.01 and HR = 7.5, P = 0.01), whereas NLRP2 and USP40 genes were significant in DFS (HR = 9.62 and P = <0.005 and HR = 5.29, P = 0.01). Mutations in NLRP2 and OBSCN showed a statistically significant co-occurrence tendency (p=0.042; Log2 Odds Ratio >3). We identified that NLRP2 gene mutations are significant in both OS and DFS. In Figure 4, NLRP2 and OBSCN gene mutations have a higher risk ratio in OS than the two clinical factors. Similarly, NLRP2 and USP40 mutations have a higher risk ratio in DFS than both clinical factors.”
(lines 407-409)
“In particular, a multivariate Cox proportional hazards model suggests that patients with NLRP2, OBSCN, and USP40 mutations have a higher mortality rate than other factors.”
Q 6: There is still no focused explanation in discussion for why survival-related genes derived from the TCGA database, rarely perform survival differences in the Korean data.
Response 6: We have added an explanation to discussion reflecting your comment (Lines 529-532).
“The limitation of the present study is that the panel was developed with the genes derived from the TCGA-KIRC database due to a lack of studies on Asians with ccRCC. ​This accounts for the different results in survival-specific genes between the Korean and the Western cohorts.”
Reviewer 2 Report
I would like to thank the authors for giving me the opportunity read their manuscript.
Although there are relevant to do these kinds of studies, this manuscript can improve in terms of direction for next studies and be clearer in terms of methods and sample size justifications/ projections. Hence consider a flow diagram to illustration the numbers used in each step clearly or have a section on sample size and sample numbers separate from protocol.
Also, the rationale of the study is clear that is to identify specific survival genes in less study cohorts such as Asian study populations. However, that the end it is unclear if this was achieved, are you saying CARD6 is unique to Asians RCCs subjects? If not is any of the 11 identified are?
Results are well presented; however, the discussion can be more focus as stated above, clear what is needed next to truly suggest a panel of indicates for this disease, that it can be used by clinical and not just be a suggestion or just hold potential but eventual will be a panel?
Or none of these is part of a panel? In other words, diagnosis or prognosis panel, used by which clinicians - urologist, pathologist, oncologist or all and how? As even for bladder cancer biomarkers are just adjuvant and not the main diagnostic tool nor prognostics tool until PSA for prostate cancer.
Please also check if some of the sentences can be rephrase to reduce similarity to other sources
Thank you,
Author Response
Point 1: Although there are relevant to do these kinds of studies, this manuscript can improve in terms of direction for next studies and be clearer in terms of methods and sample size justifications/ projections. Hence consider a flow diagram to illustration the numbers used in each step clearly or have a section on sample size and sample numbers separate from protocol.
Response 1: We have added a flowchart to figure 1 to clarify the point, and also added following description. (Figure 1, Line 213-214).
“The steps we performed from gene discovery to NGS data analysis are illustrated in Figure 1.”
Point 2: Also, the rationale of the study is clear that is to identify specific survival genes in less study cohorts such as Asian study populations. However, that the end it is unclear if this was achieved, are you saying CARD6 is unique to Asians RCCs subjects? If not is any of the 11 identified are?
Response 2: We have added an explanation to discussion reflecting your comment. (line 380-382).
If you look at Table 4, among the 11 genes, only CARD6 (OS) and BAP1 (DFS) were significantly associated with survival in Koreans. However, Table 6 shows that all 11 genes were observed to be associated with OS and/or DFS in the TCGA-KIRC (American) and RECA-EU data (European).
“Therefore in all three data, CARD6 was the only gene that showed a significant association with survival (OS) in Korean, TCGA-KIRC (American), and RECA-EU data (European).”
Point 3: Results are well presented; however, the discussion can be more focus as stated above, clear what is needed next to truly suggest a panel of indicates for this disease, that it can be used by clinical and not just be a suggestion or just hold potential but eventual will be a panel? Or none of these is part of a panel? In other words, diagnosis or prognosis panel, used by which clinicians - urologist, pathologist, oncologist or all and how? As even for bladder cancer biomarkers are just adjuvant and not the main diagnostic tool nor prognostics tool until PSA for prostate cancer.
Response 3: We have added an explanation to discussion reflecting your comment. (lines 535-538).
This study is considered as an initial step towards precision medicine in ccRCC patients. First, to create a gene panel, we need a list of genes that have been identified and validated in a large number of the patients. Second, for the gene panel to be used internationally, it is necessary to identify mutations in various races and to confirm their association with clinicopathologic findings.
“The authors expect that, through the results of this study, a gene panel that predicts the prognosis of ccRCC patients will be developed and utilized by clinicians including pathologists, urologists, oncologists and radiation oncologists. However, many challenges remain to provide a reliable prognostic strategy for ccRCC patients. “
Point 4: Please also check if some of the sentences can be rephrase to reduce similarity to other sources.
Response 4: We checked for similarity using iThenticate and obtained the result of 28% similarity in our manuscript excluding the reference as shown in the screenshot below. After reviewing the report, the similarity was observed mainly in methods and discussions. It was identified that the similarity arises from the generic terms and phrases used to describe the materials and methods in the study. Also similarities were reported in the discussion, which were quotes from the papers used as references. Therefore, we finally confirmed that there is no problem of plagiarism in our manuscript.